# Reinforcement Learning with Augmented Data

**Michael Laskin**[*]
UC Berkeley

**Kimin Lee**[*]
UC Berkeley

**Adam Stooke**
UC Berkeley

**Lerrel Pinto**
New York University

**Pieter Abbeel**
UC Berkeley

**Aravind Srinivas**
UC Berkeley

## Abstract

Learning from visual observations is a fundamental yet challenging problem in Reinforcement Learning (RL). Although algorithmic advances combined with convolutional neural networks have proved to be a recipe for success, current methods are still lacking on two fronts: (a) data-efficiency of learning and (b) generalization to new environments. To this end, we present Reinforcement Learning with Augmented Data (RAD), a simple plug-and-play module that can enhance most RL algorithms. We perform the first extensive study of general data augmentations for RL on both pixel-based and state-based inputs, and introduce two new data augmentations - *random translate* and *random amplitude scale*. We show that augmentations such as random translate, crop, color jitter, patch cutout, random convolutions, and amplitude scale can enable simple RL algorithms to outperform complex state-of-the-art methods across common benchmarks. RAD sets a new state-of-the-art in terms of data-efficiency and final performance on the DeepMind Control Suite benchmark for pixel-based control as well as OpenAI Gym benchmark for state-based control. We further demonstrate that RAD significantly improves test-time generalization over existing methods on several OpenAI ProcGen benchmarks. Our RAD module and training code are available at https://www.github.com/MishaLaskin/rad.

## 1 Introduction

Learning from visual observations is a fundamental problem in reinforcement learning (RL). Current success stories build on two key ideas: (a) using expressive convolutional neural networks (CNNs) [1] that provide strong spatial inductive bias; (b) better credit assignment [2–4] techniques that are crucial for sequential decision making. This combination of CNNs with modern RL algorithms has led to impressive success with human-level performance in Atari [2], super-human Go players [5], continuous control from pixels [3, 4] and learning policies for real-world robot grasping [6].

While these achievements are truly impressive, RL is notoriously plagued with poor data-efficiency and generalization capabilities [7, 8]. Real-world successes of reinforcement learning often require *months* of data-collection and (or) training [6, 9]. On the other hand, biological agents have the remarkable ability to learn quickly [10, 11], while being able to generalize to a wide variety of unseen tasks [12]. These challenges associated with RL are further exacerbated when we operate on pixels due to high-dimensional and partially-observable inputs. Bridging the gap of data-efficiency and generalization is hence pivotal to the real-world applicability of RL.

Supervised learning, in the context of computer vision, has addressed the problems of data-efficiency and generalization by injecting useful priors. One such often ignored prior is *Data Augmentation*. It

---

[*]Equal contribution.

was critical to the early successes of CNNs [1, 13] and has more recently enabled better supervised [14, 15], semi-supervised [16–18] and self-supervised [19–21] learning. By using multiple augmented views of the same data-point as input, CNNs are forced to learn consistencies in their internal representations. This results in a visual representation that improves generalization [17, 19–21], data-efficiency [16, 19, 20] and transfer learning [19, 21].

Inspired by the impact of data augmentation in computer vision, we present RAD: **R**einforcement Learning with **A**ugmented **D**ata, a technique to incorporate data augmentations on input observations for reinforcement learning pipelines. Through RAD, we ensure that the agent is learning on multiple views (or augmentations) of the same input (see Figure 1). This allows the agent to improve on *two key capabilities*: (a) **data-efficiency**: learning to quickly master the task at hand with drastically fewer experience rollouts; (b) **generalization**: improving transfer to unseen tasks or levels simply by training on more diversely augmented samples. To the best of our knowledge, we present the first extensive study of the use of data augmentation for reinforcement learning with *no changes* to the underlying RL algorithm and no additional assumptions about the domain other than the knowledge that the agent operates from image-based or proprioceptive (positions & velocities) observations.

We highlight the main contributions of RAD below:

- We show that *RAD outperforms prior state-of-the-art baselines* on both the widely used pixel-based DeepMind control benchmark [22] as well as state-based OpenAI Gym benchmark [23]. On both benchmark, RAD sets a new state-of-the-art *in terms data-efficiency and asymptotic performance* on the majority of environments tested.
- We show that RAD significantly *improves test-time generalization* on several environments in the OpenAI ProcGen benchmark suite [24] widely used for generalization in RL.
- We *introduce two new data augmentations*: **random translation** for image-based input and **random amplitude scaling** for proprioceptive input that are utilized to achieve state-of-the-art results. To the best of our knowledge, these augmentations were not used in prior work.

## 2 Related work

### 2.1 Data augmentation in computer vision

Data augmentation in deep learning systems for computer vision can be found as early as LeNet-5 [1], an early implementation of CNNs on MNIST digit classification. In AlexNet [13] wherein the authors applied CNNs to image classification on ImageNet [25], data augmentations, such as random flip and crop, were used to improve the classification accuracy. These data augmentations inject the priors of invariance to translation and reflection, playing a significant role in improving the performance of supervised computer vision systems. Recently, new augmentation techniques such as AutoAugment [14] and RandAugment [15] have been proposed to further improve the performance. For unsupervised and semi-supervised learning, several unsupervised data augmentation techniques have been proposed [18, 16, 26]. In particular, contrastive representation learning approaches [19–21] with data augmentations have recently dramatically improved the label-efficiency of downstream vision tasks like ImageNet classification.

### 2.2 Data augmentation in reinforcement learning

Data augmentation has also been investigated in the context of RL though, to the best of our knowledge, there was no extensive study on a variety of widely used benchmarks prior to this work. For improving generalization in RL, domain randomization [27–29] was proposed to transfer policies from simulation to the real world by utilizing diverse simulated experiences. Cobbe et al. [30] and Lee et al. [31] showed that simple data augmentation techniques such as cutout [30] and random convolution [31] can be useful to improve generalization of agents on the OpenAI CoinRun and ProcGen benchmarks.

To improve the data-efficiency, CURL [32] utilized data augmentations for learning contrastive representations in the RL setting. While the focus in CURL was to make use of data augmentations jointly through contrastive and reinforcement learning losses, RAD attempts to directly use data augmentations for reinforcement learning without any auxiliary loss (see Section I for discussions on tradeoffs between CURL and RAD). Concurrent and independent to our work, DrQ [33] utilized

random cropping and regularized Q-functions in conjunction with the off-policy RL algorithm SAC [34]. On the other hand, RAD can be plugged into any reinforcement learning method (on-policy methods like PPO [4] and off-policy methods like SAC [34]) *without making any changes to the underlying algorithm*.

For a more detailed and comprehensive discussion of prior work, we refer the reader to Appendix A.

## 3 Background

RL agents act within a Markov Decision Process, defined as the tuple $(\mathcal{S}, \mathcal{A}, P, \gamma)$, with the following components: states $s \in \mathcal{S} = \mathbb{R}^n$, actions $a \in \mathcal{A}$, and state transition distribution, $P = P(s_{t+1}, r_t | s_t, a_t)$, which defines the task mechanics and rewards. Without prior knowledge of $P$, the RL agent's goal is to use experience to maximize expected rewards, $R = \sum_{t=0}^{\infty} \gamma^t r_t$, under discount factor $\gamma \in [0, 1)$. Crucially, in RL from pixels, the agent receives image-based observations, $o_t = O(s_t) \in \mathbb{R}^k$, which are a high-dimensional, indirect representation of the state.

**Soft Actor-Critic**. SAC [34] is a state-of-the-art off-policy algorithm for continuous controls. SAC learns a policy $\pi_\psi(a|o)$ and a critic $Q_\phi(o, a)$ by maximizing a weighted objective of the reward and the policy entropy, $\mathbb{E}_{s_t, a_t \sim \pi} \left[ \sum_t r_t + \alpha \mathcal{H}(\pi(\cdot|o_t)) \right]$. The critic parameters are learned by minimizing the squared Bellman error using transitions $\tau_t = (o_t, a_t, o_{t+1}, r_t)$ from an experience buffer $\mathcal{D}$,

$$\mathcal{L}_Q(\phi) = \mathbb{E}_{\tau \sim \mathcal{D}} \left[ \left( Q_\phi(o_t, a_t) - (r_t + \gamma V(o_{t+1})) \right)^2 \right]. \tag{1}$$

The target value of the next state can be estimated by sampling an action using the current policy:

$$V(o_{t+1}) = \mathbb{E}_{a' \sim \pi} \left[ Q_{\bar{\phi}}(o_{t+1}, a') - \alpha \log \pi_\psi(a'|o_{t+1}) \right], \tag{2}$$

where $Q_{\bar{\phi}}$ represents a more slowly updated copy of the critic. The policy is learned by minimizing the divergence from the exponential of the soft-Q function at the same states:

$$\mathcal{L}_\pi(\psi) = -\mathbb{E}_{a \sim \pi} \left[ Q_\phi(o_t, a) - \alpha \log \pi_\psi(a|o_t) \right], \tag{3}$$

via the reparameterization trick for the newly sampled action. $\alpha$ is learned against a target entropy.

**Proximal policy optimization**. PPO [4] is a state-of-the-art on-policy algorithm for learning a continuous or discrete control policy, $\pi_\theta(a|o)$. PPO forms policy gradients using action-advantages, $A_t = A^\pi(a_t, s_t) = Q^\pi(a_t, s_t) - V^\pi(s_t)$, and minimizes a clipped-ratio loss over minibatches of recent experience (collected under $\pi_{\theta_{old}}$):

$$\mathcal{L}_\pi(\theta) = -\mathbb{E}_{\tau \sim \pi} \left[ \min \left( \rho_t(\theta) A_t, \text{clip}(\rho_t(\theta), 1 - \epsilon, 1 + \epsilon) A_t \right) \right], \quad \rho_t(\theta) = \frac{\pi_\theta(a_t|o_t)}{\pi_{\theta_{old}}(a_t|o_t)}. \tag{4}$$

Our PPO agents learn a state-value estimator, $V_\phi(s)$, which is regressed against a target of discounted returns and used with Generalized Advantage Estimation [4]:

$$\mathcal{L}_V(\phi) = \mathbb{E}_{\tau \sim \pi} \left[ \left( V_\phi(o_t) - V_t^{targ} \right)^2 \right]. \tag{5}$$

## 4 Reinforcement learning with augmented data

We investigate the utility of data augmentations in model-free RL for both off-policy and on-policy settings by processing image observations with stochastic augmentations before passing them to the agent for training. For the base RL agent, we use SAC [34] and PPO [4] as the off-policy and on-policy RL methods respectively. During training, we sample observations from either a replay buffer or a recent trajectory and augment the images within the minibatch. In the RL setting, it is common to stack consecutive frames as observations to infer temporal information such as object velocities. Crucially, augmentations are applied randomly across the batch but consistently across the frame stack [32] as shown in Figure 1.[2] This enables the augmentation to retain temporal information present across the frame stack.

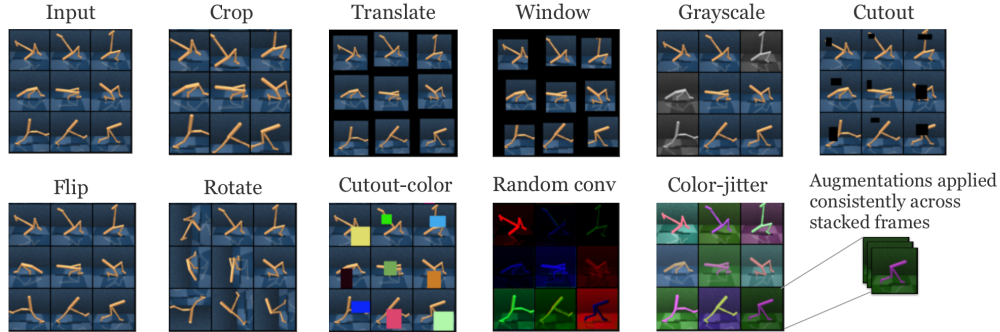

Figure 1: We investigate ten different types of data augmentations - crop, translate, window, grayscale, cutout, cutout-color, flip, rotate, random convolution, and color-jitter. During training, a minibatch is sampled from the replay buffer or a recent trajectory randomly augmented. While augmentation across the minibatch is stochastic, it is consistent across the stacked frames.

**Augmentations of image-based input:**    Across our experiments, we investigate and ablate crop, translate, window, grayscale, cutout, cutout-color, flip, rotate, random convolution, and color jitter augmentations, which are shown in Figure 1. Of these, the *translate and window are novel augmentations* that we did not encounter in prior work.

**Crop**: Extracts a random patch from the original frame. As our experiments will confirm, the intuition behind random cropping is primarily to imbue the agent with additional translation invariance. **Translate**: *random translation* renders the full image within a larger frame and translates the image randomly across the larger frame. For example, a $100 \times 100$ pixel image could be randomly translated within a $108 \times 108$ empty frame. For example, in DMControl we render $100 \times 100$ pixel frames and crop randomly to $84 \times 84$ pixels. **Window**: Selects a random window from an image by masking out the cropped part of the image. **Grayscale**: Converts RGB images to grayscale with some random probability $p$. **Cutout**: Randomly inserts a small black occlusion into the frame, which may be perceived as cutting out a small patch from the originally rendered frame. **Cutout-color**: Another variant of cutout where instead of rendering black, the occlusion color is randomly generated. **Flip**: Flips an image at random across the vertical axis. **Rotate**: Randomly samples an angle from the following set $\{0°, 90°, 180°, 270°\}$ and rotates the image accordingly. **Random convolution**: First introduced in [31], augments the image color by passing the input observation through a random convolutional layer. **Color jitter**: Converts RGB image to HSV and adds noise to the HSV channels, which results in explicit color jittering.

**Extension to state-based inputs:**    We also consider an extension to state-based inputs such as proprioceptive features (e.g., positions and velocities). Specifically, we investigate two data augmentation techniques: (a) **random amplitude scaling**, introduced in this work, multiplies the uniform random variable, i.e., $s' = s * z$, where $z \sim \text{Uni}[\alpha, \beta]$, and (b) **Gaussian noise** adds Gaussian random variable, i.e., $s' = s + z$, where $z \sim \mathcal{N}(0, I)$. Here, $s', s$ and $z$ are all vectors (see Appendix J for more details). Similar to image inputs, augmentations are applied randomly across the batch but consistently across the time, i.e., same randomization to current and next input states. Of these, *random amplitude scaling is a novel augmentation*. The intuition behind these augmentations is that *random amplitude scaling* randomizes the amplitude of input states while maintaining their intrinsic information (e.g., sign of inputs). We also remark that *Gaussian noise* is related to offset invariance.

## 5   Experimental results

### 5.1   Setup

**DMControl:** First, we focus on studying the data-efficiency of our proposed methods on pixel-based RL. To this end, we utilize the DeepMind Control Suite (DMControl) [22], which has recently become a common benchmark for comparing efficient RL agents, both model-based and model-free. DMControl presents a variety of complex tasks including bipedal balance, locomotion, contact forces,

Table 1: We report scores for RAD and baseline methods on DMControl100k and DMControl500k. In both settings, RAD achieves state-of-the-art performance on all (**6** out of **6**) environments. We selected these 6 environments for benchmarking due to availability of baseline performance data from CURL [32], PlaNet [35], Dreamer [36], SAC+AE [37], and SLAC [38]. We also show performance data on 15 environments in total in the Appendix D. Results are reported as averages across 10 seeds for the 6 main environments. A full list of hyperparameters is provided in Table 4 of Appendix E.

| 500K STEP SCORES | RAD | CURL | PLANET | DREAMER | SAC+AE | SLACv1 | PIXEL SAC | STATE SAC |
|---|---|---|---|---|---|---|---|---|
| FINGER, SPIN | **947** ± 101 | 926 ± 45 | 561 ± 284 | 796 ± 183 | 884 ± 128 | 673 ± 92 | 192 ± 166 | 923 ± 211 |
| CARTPOLE, SWING | **863** ± 9 | 845 ± 45 | 475 ± 71 | 762 ± 27 | 735 ± 63 | - | 419 ± 40 | 848 ± 15 |
| REACHER, EASY | **955** ± 71 | 929 ± 44 | 210 ± 44 | 793 ± 164 | 627 ± 58 | - | 145 ± 30 | 923 ± 24 |
| CHEETAH, RUN | **728** ± 71 | 518 ± 28 | 305 ± 131 | 570 ± 253 | 550 ± 34 | 640 ± 19 | 197 ± 15 | 795 ± 30 |
| WALKER, WALK | **918** ± 16 | 902 ± 43 | 351 ± 58 | 897 ± 49 | 847 ± 48 | 842 ± 51 | 42 ± 12 | 948 ± 54 |
| CUP, CATCH | **974** ± 12 | 959 ± 27 | 460 ± 380 | 879 ± 87 | 794 ± 58 | 852 ± 71 | 312 ± 63 | 974 ± 33 |
| 100K STEP SCORES | | | | | | | | |
| FINGER, SPIN | **856** ± 73 | 767 ± 56 | 136 ± 216 | 341 ± 70 | 740 ± 64 | 693 ± 141 | 224 ± 101 | 811 ± 46 |
| CARTPOLE, SWING | **828** ± 27 | 582 ± 146 | 297 ± 39 | 326 ± 27 | 311 ± 11 | - | 200 ± 72 | 835 ± 22 |
| REACHER, EASY | **826** ± 219 | 538 ± 233 | 20 ± 50 | 314 ± 155 | 274 ± 14 | - | 136 ± 15 | 746 ± 25 |
| CHEETAH, RUN | **447** ± 88 | 299 ± 48 | 138 ± 88 | 235 ± 137 | 267 ± 24 | 319 ± 56 | 130 ± 12 | 616 ± 18 |
| WALKER, WALK | **504** ± 191 | 403 ± 24 | 224 ± 48 | 277 ± 12 | 394 ± 22 | 361 ± 73 | 127 ± 24 | 891 ± 82 |
| CUP, CATCH | **840** ± 179 | 769 ± 43 | 0 ± 0 | 246 ± 174 | 391 ± 82 | 512 ± 110 | 97 ± 27 | 746 ± 91 |

and goal-reaching with both sparse and dense reward signals. For DMControl experiments, we evaluate the data-efficiency by measuring the performance of our method at 100k (i.e., low sample regime) and 500k (i.e., asymptotically optimal regime) *simulator or environment* steps[3] during training by following the setup in CURL [32]. These benchmarks are referred to as *DMControl100k* and *DMControl500k*. For comparison, we consider six powerful recent pixel-based methods: CURL [32] learns contrastive representations, SLAC [38] learns a forward model and uses it to shape encoder representations, while SAC+AE [37] minimizes a reconstruction loss as an auxiliary task. All three methods use SAC [34] as their base algorithm. Dreamer [36] and PlaNet [35] learn world models and use them to generate synthetic rollouts similar to Dyna [39]. Pixel SAC is a vanilla Soft Actor-Critic operating on pixel inputs, and state SAC is an *oracle* baseline that operates on the proprioceptive state of the simulated agent, which includes joint positions and velocities. We also provide learning curves for longer runs and examine how RAD compares to state SAC and CURL across a more diverse set of environments in Appendix D.

**ProcGen:** Although DMControl is suitable for benchmarking data-efficiency and performance, it evaluates the performance on the same environment in which the agent was trained and is thus not applicable for studying generalization. For this reason, we focus on the OpenAI ProcGen benchmarks [24] to investigate the generalization capabilities of RAD. ProcGen presents a suite of game-like environments where the train and test environments differ in visual appearance and structure. For this reason, it is a commonly used benchmark for studying the generalization abilities of RL agents [30]. Specifically, we evaluate the zero-shot performance of the trained agents on the full distribution of unseen levels. Following the setup in ProcGen [24], we use the CNN architecture found in IMPALA [40] as the policy network and train the agents using the Proximal Policy Optimization (PPO) [4] algorithm for 20M timesteps. For all experiments, we use the *easy environment difficulty* and the hyperparameters suggested in [24], which have been shown to be empirically effective.

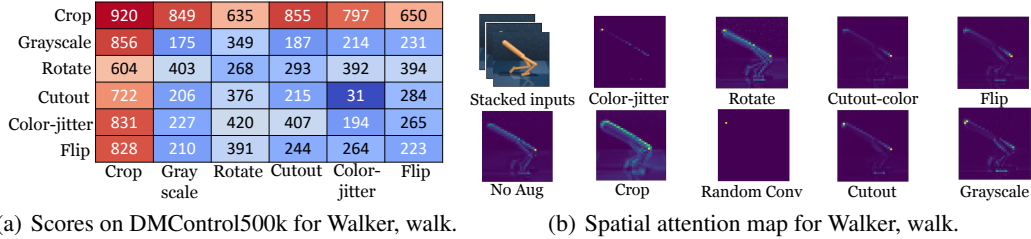

| | Crop | Gray scale | Rotate | Cutout | Color-jitter | Flip |
|---|---|---|---|---|---|---|
| Crop | 920 | 849 | 635 | 855 | 797 | 650 |
| Grayscale | 856 | 175 | 349 | 187 | 214 | 231 |
| Rotate | 604 | 403 | 268 | 293 | 392 | 394 |
| Cutout | 722 | 206 | 376 | 215 | 31 | 284 |
| Color-jitter | 831 | 227 | 420 | 407 | 194 | 265 |
| Flip | 828 | 210 | 391 | 244 | 264 | 223 |

(a) Scores on DMControl500k for Walker, walk.    (b) Spatial attention map for Walker, walk.

Figure 2: (a) We ablate six common data augmentations on the walker, walk environment by measuring performance on DMControl500k of each permutation of any two data augmentations being performed in sequence. For example, the *crop* row and *grayscale* column correspond to the score achieved after applying random crop and then random grayscale to the input images (entries along the main axis use only one application of the augmentation). (b) Spatial attention map of an encoder that shows where the agent focuses on in order to make a decision in Walker Walk environment. Random crop enables the agent to focus on the robot body and ignore irrelevant scene details compared to other augmentations as well as the base agent that learns without any augmentation.

**OpenAI Gym.** For OpenAI Gym experiments with proprioceptive inputs (e.g., positions and velocities), we compare to PETS [41], a model-based RL algorithm that utilizes ensembles of dynamics models; POPLIN-P [42], a state-of-the-art model-based RL algorithm which uses a policy network to generate actions for planning; POPLIN-A [42], variant of POPLIN-P which adds noise in the action space; METRPO [43], model-based policy optimization based on TRPO [44]; and two state-of-the-art model-free algorithms, TD3 [45] and SAC [34]. In our experiments, we apply RAD to SAC. Following the experimental setups in POPLIN [42], we report the mean and standard deviation across four runs on Cheetah, Walker, Hopper, Ant, Pendulum and Cartpole environments.

## 5.2 Improving data-efficiency on DeepMind Control Suite

**Data-efficiency:** Mean scores shown in Table 1 and learning curves in Figure 7 show that data augmentation significantly improves the data-efficiency and performance across the six extensively benchmarked environments compared to existing methods. We summarize the main findings below:

- RAD is the **state-of-the-art algorithm** on all (**6** out of **6**) environments on both DMControl100k and DMControl500k benchmarks.
- RAD **improves** the performance of **pixel SAC by 4x** on both DMControl100k and DMControl500k *solely through data augmentation* without learning a forward model or any other auxiliary task.
- RAD **matches** the performance of **state-based SAC** on the majority of (**11** out of **15**) DMControl environments tested as shown in Figure 7.
- **Random translation** or **random crop**, stand-alone, have the highest impact on final performance relative to all other augmentations as shown in Figure 2(a).

**Which data augmentations are the most effective?** To answer this question for DMControl, we ran RAD with permutations of two data augmentations applied in sequence (e.g., crop followed by grayscale) on the *Walker Walk* environment and report the scores at 500k environment steps. We chose this environment because SAC without augmentation fails at this task, resulting in scores well below 200. Our results, shown in Figure 2(a), indicate that most data augmentations improve the performance of the base policy, and that random crop by itself was the most effective by a large margin.

**Why is random crop so effective?** To analyze the effects of random crop, we decompose it into its two component augmentations: (a) *random window*, which masks a random boundary region of the image, exactly where crop would remove it, and (b) *random translate*, which places the full image entirely within a larger frame of zeros, but at a random location. In Appendix C, Figure 6 shows resulting learning curves from each augmentation. The benefit of translations is clearly demonstrated, whereas the random information hiding due to windowing produced little effect. Table 1 reports scores using *random translate*, a new SOTA method, for all environments except for *Walker Walk*,

Table 2: We present the generalization results of RAD with different data augmentation methods on the three OpenAI ProcGen environments: BigFish, StarPilot and Jumper. We report the test performances after 20M timesteps. The results show the mean and standard deviation averaged over three runs. We see that RAD is able to outperform the baseline PPO trained on two times the number of training levels benefitting from data augmentations such as random crop, cutout and color jitter.

| | # of training levels | Pixel PPO | RAD (gray) | RAD (flip) | RAD (rotate) | RAD (random conv) | RAD (color-jitter) | RAD (cutout) | RAD (cutout-color) | RAD (crop) |
|---|---|---|---|---|---|---|---|---|---|---|
| BigFish | 100 | 1.9 ± 0.1 | 1.5 ± 0.3 | 2.3 ± 0.4 | 1.9 ± 0.0 | 1.0 ± 0.1 | 1.0 ± 0.1 | 2.9 ± 0.2 | 2.0 ± 0.2 | **5.4** ± 0.5 |
| | 200 | 4.3 ± 0.5 | 2.1 ± 0.3 | 3.5 ± 0.4 | 1.5 ± 0.6 | 1.2 ± 0.1 | 1.5 ± 0.2 | 3.3 ± 0.2 | 3.5 ± 0.3 | **6.7** ± 0.8 |
| StarPilot | 100 | 18.0 ± 0.7 | 10.6 ± 1.4 | 13.1 ± 0.2 | 9.7 ± 1.6 | 7.4 ± 0.7 | 15.0 ± 1.1 | 17.2 ± 2.0 | **22.4** ± 2.1 | 20.3 ± 0.7 |
| | 200 | 20.3 ± 0.7 | 20.6 ± 1.0 | 20.7 ± 3.9 | 15.7 ± 0.7 | 11.0 ± 1.5 | 20.6 ± 1.1 | 24.5 ± 0.1 | **24.5** ± 1.6 | 24.3 ± 0.1 |
| Jumper | 100 | 5.2 ± 0.5 | 5.2 ± 0.1 | 5.2 ± 0.7 | 5.7 ± 0.6 | 5.5 ± 0.3 | **6.1** ± 0.2 | 5.6 ± 0.1 | 5.8 ± 0.6 | 5.1 ± 0.2 |
| | 200 | **6.0** ± 0.2 | 5.6 ± 0.1 | 5.4 ± 0.3 | 5.5 ± 0.1 | 5.2 ± 0.1 | 5.9 ± 0.1 | 5.4 ± 0.1 | 5.6 ± 0.4 | 5.2 ± 0.7 |

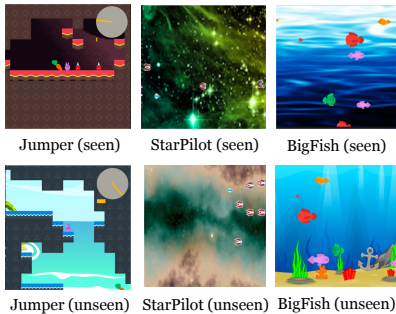

(a) ProcGen

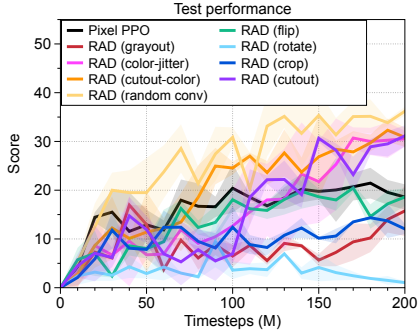

(b) Test performance on modified CoinRun

Figure 3: (a) Examples of seen and unseen environments on ProcGen. (b) The test performance under the modified CoinRun. The solid/dashed lines and shaded regions represent the mean and standard deviation, respectively.

where random crop sometimes reduced variance. In Figure 5 of Appendix C, we ablate the size of the final translation image, finding in some cases that random placement within as little as two additional pixels in height and width is sufficient to reap the benefit.

**What are the effects on the learned encoding?** To understand how the augmentations affect learned representations encoder, we visualize a spatial attention map from the output of the last convolutional layer. Similar to [46], we compute the spatial attention map by mean-pooling the absolute values of the activations along the channel dimension and follow with a 2-dimensional spatial softmax. Figure 4 visualizes the spatial attention maps for the augmentations considered. Without augmentation, the activation is highly concentrated at the point of the forward knee, whereas with random crop/translate, entire edges of the body are prominent, providing a more complete and robust representation of the state.

## 5.3 Improving generalization on OpenAI ProcGen

**Generalization:** We evaluate the generalization ability on three environments from OpenAI Procgen: BigFish, StarPilot, and Jumper (see Figure 3(a)) by varying the number of training environments and ablating for different data augmentation methods. We summarize our findings below:

Table 3: Performance on OpenAI Gym. The training timestep varies from 50,000 to 200,000 depending on the difficulty of the tasks. The results show the mean and standard deviation averaged over four runs and the best results are indicated in bold. For baseline methods, we report the best number in POPLIN [42].

| | Cheetah | Walker | Hopper | Ant | Pendulum | Cartpole |
|---|---|---|---|---|---|---|
| PETS | $2288.4 \pm 1019.0$ | $282.5 \pm 501.6$ | $114.9 \pm 621.0$ | $1165.5 \pm 226.9$ | $155.7 \pm 79.3$ | $\mathbf{199.6 \pm 4.6}$ |
| POPLIN-A | $1562.8 \pm 1136.7$ | $-105.0 \pm 249.8$ | $202.5 \pm 962.5$ | $1148.4 \pm 438.3$ | $\mathbf{178.3 \pm 19.3}$ | $200.6 \pm 1.3$ |
| POPLIN-P | $4235.0 \pm 1133.0$ | $597.0 \pm 478.8$ | $2055.2 \pm 613.8$ | $\mathbf{2330.1 \pm 320.9}$ | $167.9 \pm 45.9$ | $200.8 \pm 0.3$ |
| METRPO | $2283.7 \pm 900.4$ | $-1609.3 \pm 657.5$ | $1272.5 \pm 500.9$ | $282.2 \pm 18.0$ | $174.8 \pm 6.2$ | $138.5 \pm 63.2$ |
| TD3 | $3015.7 \pm 969.8$ | $-516.4 \pm 812.2$ | $1816.6 \pm 994.8$ | $870.1 \pm 283.8$ | $168.6 \pm 12.7$ | $-409.2 \pm 928.8$ |
| SAC | $4035.7 \pm 268.0$ | $-382.5 \pm 849.5$ | $2020.6 \pm 692.9$ | $836.5 \pm 68.4$ | $162.1 \pm 12.3$ | $\mathbf{199.8 \pm 1.9}$ |
| RAD | $\mathbf{4554.3 \pm 1209.0}$ | $\mathbf{806.4 \pm 706.7}$ | $\mathbf{2149.1 \pm 249.9}$ | $1150.9 \pm 494.6$ | $167.4 \pm 9.7$ | $199.9 \pm 0.8$ |
| Timesteps | 200000 | 200000 | 200000 | 200000 | 50000 | 50000 |

- As shown in Table 2, various data augmentation methods such as random crop and cutout **significantly improve the generalization performance** on the BigFish and StarPilot environments (see Appendix F for learning curves).

- In particular, **RAD with random crop** achieves **55.8%** relative gain over pixel-based PPO on the BigFish environment.

- RAD trained with **100** training levels outperforms the pixel-based PPO trained with **200** training levels on both BigFish and StarPilot environments. *This shows that data augmentation can be more effective in learning generalizable representations compared to simply increasing the number of training environments.*

- In the case of Jumper (a navigation task), the gain from data augmentation is not as significant because the task involves structural generalization to different map layouts and is likely to require recurrent policies [24].

- To verify the effects of data augmentations on such environments, we consider a modified version of CoinRun [30] which corresponds to a simpler version of Jumper. By following the set up in [31], we train agents on a fixed set of 500 levels with half of the available themes (style of backgrounds, floors, agents, and moving obstacles) and then measure the test performance on 1000 different levels consisting of unseen themes to evaluate the generalization ability across the visual changes. As shown in Figure 3(b), data augmentation methods, such as random convolution, color-jitter, and cutout-color, improve the generalization ability of the agent to a greater extent than random crop suggesting the need to further study data augmentations in these environments.

## 5.4 Improving state-based RL on OpenAI Gym

**Data-efficiency on state-based inputs**. Table 3 shows the average returns of evaluation rollouts for all methods (see Figure 12 in Appendix J for learning curves). We report results for state-based RAD using the best performing augmentation - random amplitude scaling; for details regarding performance of other augmentations we refer the reader to Appendix J. Similar to data augmentation in the visual setting, RAD is the state-of-the-art algorithm on the majority (**4** out of **6**) of benchmarked environments. RAD consistently improves the performance of SAC across all environments, and outperforms a competing state-of-the-art method - POPLIN-P - on most of the environments. It is worth noting that RAD improves the average return compared to POPLIN-P by **1.7x** in Walker, an environment where most prior RL methods fail, including both model-based and model-free ones. We hypothesize that random amplitude scaling is effective because it forces the agent to be robust to input noise while maintaining the intrinsic information of the state, such as sign of inputs and relative differences between them.

These results showcase the generality of incorporating inductive biases through augmentation (e.g., amplitude invariance) by showing that improving RL with data augmentation is not specific to pixel-based inputs but also applies RL from state. By achieving state-of-the-art performance across both visual and proprioceptive inputs, RAD sets a powerful new baseline for future algorithms.

## 6  Conclusion

In this work, we proposed RAD, a simple plug-and-play module to enhance any reinforcement learning (RL) method using data augmentations. For the first time, we show that data augmentations *alone* can significantly improve the data-efficiency and generalization of RL methods operating from pixels, *without any changes to the underlying RL algorithm*, on the DeepMind Control Suite and the OpenAI ProcGen benchmarks respectively. Our implementation is simple, efficient, and has been open-sourced. We hope that the performance gains, implementation ease, and wall clock efficiency of RAD make it a useful module for future research in data-efficient and generalizable RL methods; and a useful tool for facilitating real-world applications of RL.

## 7  Broader Impact

While there has been a trend in growing complexity and compute requirements to achieve state-of-the-art results in Computer Vision [20], NLP [47], and RL [48], there are two negative long-term consequences of these trends: (i) the energy demands of these large models are harmful to the environment due to increased carbon emissions if not powered by renewable energy sources (ii) they make AI research inaccessible to researchers without access to tremendous compute and engineering resources. RAD shows that, by incorporating powerful inductive biases, state-of-the-art results can be achieved with simpler methods that require less compute and model complexity than complex competing methods. RAD is therefore accessible to a broad range of researchers (even those without access to GPUs) and leaves a much smaller carbon footprint than competing methods.

While it's fair to say that even with the result from this paper, we are far removed from making Deep RL practical for solving real-world-complexity robotics problems, we believe this work provides progress towards that goal. Robots being able to learn through RL in the real world opens up opportunities for better elderly care, autonomous cleaning and disinfecting, more reliable / resilient supply chain and manufacturing operations (especially when humans might not be available due to a pandemic). On the flipside, an RL agent will optimize whatever reward one specifies. If the person in charge of the system specifies a reward that's bad for the world (or perhaps mistakenly even for themselves), the more powerful the RL, the worse the outcome. For this reason, in addition to developing better algorithms that achieve new state-of-the-art performance, it is also important to pursue complementary research on safety [49, 50].

## Acknowledgments and Disclosure of Funding

This work was supported in part by Berkeley Deep Drive (BDD), ONR PECASE N000141612723, DARPA through the LwLL program, and the Open Philanthropy Foundation.

## Footnotes

[2]For on-policy RL methods such as PPO, we apply the different augmentations across the batch but consistently across time.

[3]*environment steps* refers to the number of times the underlying simulator is stepped through. This measure is independent of policy heuristics such as action repeat. For example, if action repeat is set to 4, then 100k *environment* steps corresponds to 25k *policy* steps.

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
