[Supplementary Material · RAD_NeurIPS_2020_appendix.pdf]

# A  Extended related work

## A.1  Data augmentation in supervised learning

Since our focus is on image-based observations, we cover the related work in computer vision. Data augmentation in deep learning systems for computer vision can be found as early as LeNet-5 [1], an early implementation of CNNs on MNIST digit classification. In AlexNet [13] wherein the authors applied CNNs to image classification on ImageNet, data augmentations were used to increase the size of the original dataset by a factor of 2048 by randomly flipping and cropping $224 \times 224$ patches from the original image. These data augmentations inject the priors of invariance to translation and reflection, playing a significant role in improving the performance of supervised computer vision systems. Recently, new augmentation techniques such as AutoAugment [14] and RandAugment [15] have been proposed to further improve the performance of these systems.

## A.2  Data augmentation for data-efficiency in semi & self-supervised learning

Aside from improving supervised learning, data augmentation has also been widely utilized for unsupervised and semi-supervised learning. MixMatch [18], FixMatch [26], UDA [16] use unsupervised data augmentation in order to maximize label agreement without access to the actual labels. Several contrastive representation learning approaches [19, 21, 20] have recently dramatically improved the label-efficiency of downstream vision tasks like ImageNet classification. Contrastive approaches utilize data augmentations and perform patch-wise [19] or instance discrimination (MoCo, SimCLR) [21, 20]. In the instance discrimination setting, the contrastive objective aims to maximize agreement between augmentations of the same image and minimize it between all other images in the dataset [20, 21]. The choice of augmentations has a significant effect on the quality of the learned representations as demonstrated in SimCLR [20].

## A.3  Prior work in reinforcement learning related to data augmentation

### A.3.1  Data augmentation with domain knowledge

While not directly known for data augmentation in reinforcement learning, the following ideas can be viewed as techniques to diversify the data used to train an RL agent:

**Domain randomization** [27–29] is a simple data augmentation technique primarily used for transferring policies from simulation to the real world where one takes advantage of the simulator's access to information about the rendering and physics and thus can train transferable policies from diverse simulated experiences.

**Hindsight experience replay** [51] applies the idea of re-labeling trajectories with terminal states as fictitious goals, improving the ability of goal-conditioned RL to learn quickly with sparse rewards. This, however, makes assumptions about the goal space matching with the state space and has had limited success with pixel-based observations.

### A.3.2  Synthetic rollouts using a learned world model

While usually not viewed as a data augmentation technique, the idea of generating fake or synthetic rollouts to improve the data-efficiency of RL agents has been proposed in the Dyna framework [39]. In recent times, these ideas have been used to improve the performance of systems that have explicitly learned world models of the environment and generated synthetic rollouts using them [52, 11, 36].

### A.3.3  Data augmentation for data-efficient reinforcement learning

Data augmentation is a key component for learning contrastive representations in the RL setting as shown in the CURL framework [32], which learns representations that improve the data-efficiency of pixel-based RL by enforcing consistencies between an image and its augmented version through instance contrastive losses. Prior to our work, CURL was the state-of-the-art model for data-efficient RL from pixel inputs. While the focus in CURL was to make use of data augmentations jointly through contrastive and reinforcement learning losses, RAD attempts to directly use data augmentations for reinforcement learning without any auxiliary loss. We refer the reader to a discussion on tradeoffs between CURL and RAD in Section I. Concurrent and independent to our work, DrQ [33] uses data

augmentations and weighted Q-functions in conjunction with the off-policy RL algorithm SAC [34] to achieve state-of-the-art data-efficiency results on the DeepMind Control Suite. On the other hand, RAD can be plugged into any reinforcement learning method (on-policy methods like PPO [4] and off-policy methods like SAC [34]) *without making any changes to the underlying algorithm*. We further demonstrate the benefits of data augmentation to generalization on the OpenAI ProcGen benchmarks in addition to data-efficiency on the DeepMind Control Suite.

### A.4 Data augmentation for generalization in reinforcement learning

Cobbe et al. [30] and Lee et al. [31] showed that simple data augmentation techniques such as cutout [30] and random convolution [31] can be useful to improve generalization of agents on the OpenAI CoinRun and ProcGen benchmarks. In this paper, we extensively investigate more data augmentation techniques such as random crop and color jitter on a more diverse array of tasks. With our efficient implementation of these augmentations, we demonstrate their utility with the on-policy RL algorithm PPO [4] for the first time.

## B    Attention maps for various data augmentations

Figure 4: Spatial attention map of an encoder that shows where the agent focuses on in order to make a decision in (a) Walker Walk and (b) Cheetah Run environments. Random crop enables the agent to focus on the robot body and ignore irrelevant scene details compared to other augmentations as well as the base agent that learns without any augmentation. In addition to the agent, the base cheetah encoder focuses on the stars in the background, which are irrelevant to the task and likely harm the agent's performance. Random crop enables the encoder to capture the agent's state *much more clearly* compared to other augmentations. The quality of the attention map with random crop suggests that RAD improves the *contingency-awareness* of the agent (recognizing aspects of the environment that are under the agent's control) thereby improving its data-efficiency.

## C    Random translate ablations

Figure 5: Ablations for different sizes of the larger frame in which the image is randomly translated. The original image is always rendered at a resolution of 100x100 pixels, and then translated within a larger frame of size 102,104,108, and 116. We note that augmentation significantly improves performance compared to SAC with no augmentation on Cartpole and Cheetah environments even for the smallest frame size of 102 where minimal augmentation is happening.

Figure 6: We ablate random translation, cropping, and windowing. Since cropping can be viewed as a simultaneous translation and window operation, we wish to understand which component is responsible for the most gains. We find that the gains come primarily from the translation operation and that, on most environments, RAD with translation results in more stable and efficient learning.

# D  Learning curves for RAD on DMControl

Figure 7: We benchmark the performance of RAD relative to the best performing pixel-based baseline (CURL) as well as SAC operating on state input on 15 environments in total. RAD matches state SAC performance on the majority (**11** out of **15** environments) and performs comparably or better than CURL on all of the environments tested. Results are average values across 3 seeds.

# E   Implementation details for DMControl

For DMControl experiments, we utilize the same encoder architecture as in [32] which is similar to the architecture in [37]. We show a full list of hyperparameters for DMControl experiments in Table 4.

Table 4: Hyperparameters used for DMControl experiments. Most hyperparameters values are unchanged across environments with the exception for action repeat, learning rate, and batch size.

| Hyperparameter | Value |
|---|---|
| Augmentation | Crop - walker, walk; Translate - otherwise |
| Observation rendering | $(100, 100)$ |
| Observation down/upsampling | $(84, 84)$ (crop); $(108, 108)$ (translate) |
| Replay buffer size | 100000 |
| Initial steps | 1000 |
| Stacked frames | 3 |
| Action repeat | 2 finger, spin; walker, walk |
| | 8 cartpole, swingup |
| | 4 otherwise |
| Hidden units (MLP) | 1024 |
| Evaluation episodes | 10 |
| Optimizer | Adam |
| $(\beta_1, \beta_2) \to (f_\theta, \pi_\psi, Q_\phi)$ | $(.9, .999)$ |
| $(\beta_1, \beta_2) \to (\alpha)$ | $(.5, .999)$ |
| Learning rate $(f_\theta, \pi_\psi, Q_\phi)$ | $2e-4$ cheetah, run |
| | $1e-3$ otherwise |
| Learning rate $(\alpha)$ | $1e-4$ |
| Batch Size | 512 |
| $Q$ function EMA $\tau$ | 0.01 |
| Critic target update freq | 2 |
| Convolutional layers | 4 |
| Number of filters | 32 |
| Non-linearity | ReLU |
| Encoder EMA $\tau$ | 0.05 |
| Latent dimension | 50 |
| Discount $\gamma$ | .99 |
| Initial temperature | 0.1 |

# F   Implementation details and additional results for ProcGen

## F.1   Environment descriptions

**BigFish**. In this environment, the agent starts as a small fish and the goal is to eat fish smaller than itself. The agent can receive a small reward for eating fish and a large reward is given when it becomes bigger that all other fish. The spawn timing, position of all fish, and style of background change throughout the level.

**StarPilot**. A simple side scrolling shooter game, where the agent receive the reward by avoiding enemy. The spawn timing of all enemies and obstacles, along with their corresponding types, are changing throughout the level.

**Jumper**. An open world environment, where the goal is to find the carrot which is randomly located in the map. Style of background, location of enemy and map structure are changing throughout the level.

**Modified CoinRun**. In this task, an agent is located at the leftmost side of the map and the goal is to collect the coin located at the rightmost side of the map within 1,000 timesteps. The agent observes its surrounding environment in the third-person point of view, where the agent is always located at the center of the observation. Similar to [31], half of the available themes are utilized (i.e., style of backgrounds, floors, agents, and moving obstacles) for training.

## F.2 Implementation details

For ProcGen experiments, we follow the hyperparameters proposed in [24], which are empirically shown to be effective. Specifically, we use the CNN architecture found in IMPALA [40] as the policy network, and train the agents using the Proximal Policy Optimization (PPO) with following hyperparameters:

Table 5: Hyperparameters used for ProcGen experiments.

| Hyperparameter | Value |
| --- | --- |
| Observation rendering | $(64, 64)$ |
| Discount $\gamma$ | .99 |
| GAE parameter $\lambda$ | 0.95 |
| # of timesteprs per rollout | 256 |
| # of minibatches per rollout | 8 |
| Entropy bonus | 0.1 |
| PPO clip range | 0.2 |
| Reward Normalization | Yes |
| # of Workers | 1 |
| # of environments per worker | 64 |
| Total timesteps | 20M |
| LSTM | No |
| Frame Stack | No |
| Optimizer | Adam |
| Learning rate $(\alpha)$ | $5e-4$ |

## F.3 Learning curves

| (a) Train (100) | (b) Test (100) | (c) Train (200) | (d) Test (200) |

Figure 8: Learning curves of PPO and RAD agents trained with (a/b) 100 and (c/d) 200 training levels on StarPilot. The solid line and shaded regions represent the mean and standard deviation, respectively, across three runs.

| (a) Train (100) | (b) Test (100) | (c) Train (200) | (d) Test (200) |

Figure 9: Learning curves of PPO and RAD agents trained with (a/b) 100 and (c/d) 200 training levels on Bigfish. The solid line and shaded regions represent the mean and standard deviation, respectively, across three runs.

| (a) Train (100) | (b) Test (100) | (c) Train (200) | (d) Test (200) |

Figure 10: Learning curves of PPO and RAD agents trained with (a/b) 100 and (c/d) 200 training levels on Jumper. The solid line and shaded regions represent the mean and standard deviation, respectively, across three runs.

Figure 11: Learning curves of PPO and RAD in the modified Coinrun. The solid line and shaded regions represent the mean and standard deviation, respectively, across three runs.

## G   Code for select augmentations

```python
def random_crop(imgs, size=84):
    n, c, h, w = imgs.shape
    w1 = torch.randint(0, w - size + 1, (n,))
    h1 = torch.randint(0, h - size + 1, (n,))
    cropped = torch.empty((n, c, size, size),
        dtype=imgs.dtype, device=imgs.device)
    for i, (img, w11, h11) in enumerate(zip(imgs, w1, h1)):
        cropped[i][:] = img[:, h11:h11 + size, w11:w11 + size]
    return cropped

def random_cutout(imgs, min_cut=4, max_cut=24):
    n, c, h, w = imgs.shape
    w_cut = torch.randint(min_cut, max_cut + 1, (n,)) # random size cut
    h_cut = torch.randint(min_cut, max_cut + 1, (n,)) # rectangular shape
    fills = torch.randint(0, 255, (n, c, 1, 1)) # assume uint8.
    for img, wc, hc, fill in zip(imgs, w_cut, h_cut, fills):
        w1 = torch.randint(w - wc + 1, ()) # uniform over interior
        h1 = torch.randint(h - hc + 1, ())
        img[:, h1:h1 + hc, w1:w1 + wc] = fill
    return imgs

def random_flip(imgs, p=0.5):
    n, _, _, _ = imgs.shape
    flip_mask = torch.rand(n, device=imgs.device) < p
    imgs[flip_mask] = imgs[flip_mask].flip([3]) # left-right
    return imgs
```

# H  Time-efficiency of data augmentation

The primary gain of our data augmentation modules is enabling efficient augmentation of stacked frame inputs in the minibatch setting. Since the augmentations must be applied randomly across the batch but consistently across the frame stack, traditional frameworks like Tensorflow and PyTorch that focus on augmenting single-frame static datasets, are unsuitable for this task. We further show wall-clock efficiency relative to the PyTorch API in Table 6.

Table 6: We compare the data augmentation speed between the RAD augmentation modules and performing the same augmentations in PyTorch. We calculate the number of additional minutes required to perform 100k training steps. On average, the RAD augmentations are nearly 2x faster than augmentations accessed through the native PyTorch API. Additionally, since the PyTorch API is meant for processing single-frame images, it is not designed to apply augmentations consistently across the frame stack but randomly across the batch. Cutout and random convolution augmentations are not present in the PyTorch API.

|  | OURS | PYTORCH |
|---|---|---|
| CROP | 31.8 | 33.5 |
| GRAYSCALE | 15.6 | 51.2 |
| CUTOUT | 36.6 | - |
| CUTOUT COLOR | 45.2 | - |
| FLIP | 4.9 | 37.0 |
| ROTATE | 46.5 | 62.4 |
| RANDOM CONV. | 45.8 | - |

# I  Discussion

## I.1  CURL vs RAD

Both CURL and RAD improve the data-efficiency of RL agents by enforcing consistencies in the input observations presented to the agent. CURL does this with an explicit instance contrastive loss between an image and its augmented version using the MoCo [21] mechanism. On the other hand, RAD does not employ any auxiliary loss and directly trains the RL objective on multiple augmented views of the observations, thereby ensuring consistencies on the augmented views implicitly. The performance of RAD matches that of CURL and surpasses CURL on some of the environments in the DeepMind Control Suite (refer to Figure 7). This suggests the potential conclusion that data augmentation is sufficient for data-efficient reinforcement learning from pixels. We argue that the conclusion requires a bit more nuance in the following subsection.

## I.2  Is data augmentation sufficient for RL from pixels?

The improved performance of RAD over CURL can be attributed to the following line of thought: While both methods try to improve the data-efficiency through augmentation consistencies (CURL explicitly, RAD implicitly); RAD outperforms CURL because *it only optimizes for what we care about, which is the task reward*. CURL, on the other hand, jointly optimizes the reinforcement and contrastive learning objectives. If the metric used to evaluate and compare these methods is the score attained on the task at hand, a method that purely focuses on reward optimization is expected to be better as long as it implicitly ensures similarity consistencies on the augmented views (in this case, just by training the RL objective on different augmentations directly).

However, we believe that a representation learning method like CURL is *arguably* a more general framework for the usage of data augmentations in reinforcement learning. CURL can be applied *even without any task (or environment) reward* available. The contrastive learning objective in CURL that ensures consistencies between augmented views is disentangled from the reward optimization (RL) objective and is therefore capable of learning-rich semantic representations from high dimensional observations gathered from random rollouts. Real-world applications of RL might involve performing plenty of interactions (or rollouts) with sparse reward signals, and tasks presented to the agent as image-based goals. In such scenarios, CURL and other representation learning methods are *likely*

to be more important even though current RL benchmarks are primarily about single or multi-task reward optimization.

Given these subtle considerations, we believe that both RAD and representation learning methods like CURL will be useful tools for an RL practitioner in future research encompassing data-efficient and generalizable RL.

## J   Implementation details and additional results for OpenAI Gym

### J.1   Implementation details

We consider a combination of SAC and RAD using the publicly released implementation repository (`https://github.com/vitchyr/rlkit`) without any modifications on hyperparameters and architectures. For random amplitude scaling, we consider two variants: (a) random amplitude scaling with a single variable (RAS-S) that multiplies the one-dimensional uniform random variable, i.e., $\mathbf{s}' = \mathbf{s}*z$, where $z \sim \text{Uni}[\alpha, \beta]$, and (b) random amplitude scaling with a multivariate variable (RAS-M) that multiplies the multivariate uniform random variable, i.e., $\mathbf{s}' = \mathbf{s} * \mathbf{z}$, where $\mathbf{z} \sim \text{Uni}[\alpha, \beta]$. The minimum value of uniform distribution is chosen from $\alpha \in \{0.6, 0.8\}$ and the maximum value of uniform distribution is chosen from $\beta \in \{1.2, 1.4\}$. For Gaussian noise, we add Gaussian random variable, i.e., $\mathbf{s}' = \mathbf{s} + \mathbf{z}$, where $\mathbf{z} \sim \mathcal{N}(0, I)$. The optimal parameters are chosen to achieve the best performance on training environments.

### J.2   Experimental results on OpenAI Gym

As shown in Table 7 and Figure 12, random amplitude scaling is effective in almost all environments. We expect that this is because random amplitude scaling forces the agent to be robust to input noise while maintaining the intrinsic information of the state. However, in the case of random amplitude scaling with multiple variables, the relative differences can be changed. Because of that, random amplitude with a single scalar achieves the better performance on most environments. We also remark that a simple normalization technique such as batch normalization is not very effective compared to RAD, which implies that the gains from RAD can not be achieved by normalization.

Table 7: Performance of variants of RAD, i.e., random amplitude scaling with a single variable (RAS-S), random amplitude scaling with a multivariate variable (RAS-M), and Gaussian noise (GN), on OpenAI Gym. The training timestep varies from 50,000 to 200,000 depending on the difficulty of the tasks. The results show the mean and standard deviation averaged over four runs and the best results are indicated in bold. For baseline methods, we report the best number in POPLIN [42]. For TD3, we remark that similar scores can be reproduced by the official codebase from the POPLIN paper (e.g., 3273.4 on Cheetah and -447.3 on Walker) using 10 random seeds.

| | Cheetah | Walker | Hopper |
|---|---|---|---|
| PETS | $2288.4 \pm 1019.0$ | $282.5 \pm 501.6$ | $114.9 \pm 621.0$ |
| POPLIN-A | $1562.8 \pm 1136.7$ | $-105.0 \pm 249.8$ | $202.5 \pm 962.5$ |
| POPLIN-P | $4235.0 \pm 1133.0$ | $597.0 \pm 478.8$ | $2055.2 \pm 613.8$ |
| METRPO | $2283.7 \pm 900.4$ | $-1609.3 \pm 657.5$ | $1272.5 \pm 500.9$ |
| TD3 | $3015.7 \pm 969.8$ | $-516.4 \pm 812.2$ | $1816.6 \pm 994.8$ |
| SAC | $4035.7 \pm 268.0$ | $-382.5 \pm 849.5$ | $2020.6 \pm 692.9$ |
| SAC + BN | $3386.9 \pm 549.3$ | $751.2 \pm 1017.9$ | $1854.1 \pm 329.3$ |
| SAC + RAD (RAS-S) | $\mathbf{4554.3 \pm 1209.0}$ | $370.4 \pm 579.3$ | $\mathbf{2149.1 \pm 249.9}$ |
| SAC + RAD (RAS-M) | $2787.4 \pm 466.8$ | $\mathbf{806.4 \pm 706.7}$ | $2096.1 \pm 442.7$ |
| SAC + RAD (GN) | $2222.3 \pm 418.3$ | $-121.6 \pm 664.6$ | $19.0 \pm 619.6$ |
| Timesteps | 200000 | 200000 | 200000 |

| | Ant | Pendulum | Cartpole |
|---|---|---|---|
| PETS | $1165.5 \pm 226.9$ | $155.7 \pm 79.3$ | $\mathbf{199.6 \pm 4.6}$ |
| POPLIN-A | $1148.4 \pm 438.3$ | $\mathbf{178.3 \pm 19.3}$ | $\mathbf{200.6 \pm 1.3}$ |
| POPLIN-P | $\mathbf{2330.1 \pm 320.9}$ | $167.9 \pm 45.9$ | $\mathbf{200.8 \pm 0.3}$ |
| METRPO | $282.2 \pm 18.0$ | $174.8 \pm 6.2$ | $138.5 \pm 63.2$ |
| TD3 | $870.1 \pm 283.8$ | $168.6 \pm 12.7$ | $-409.2 \pm 928.8$ |
| SAC | $836.5 \pm 68.4$ | $162.1 \pm 12.3$ | $\mathbf{199.8 \pm 1.9}$ |
| SAC + BN | $580.8 \pm 70.2$ | $158.4 \pm 10.2$ | $178.1 \pm 38.2$ |
| SAC + RAD (RAS-S) | $1150.9 \pm 494.6$ | $158.9 \pm 16.8$ | $\mathbf{199.9 \pm 0.8}$ |
| SAC + RAD (RAS-M) | $966.6 \pm 321.0$ | $167.4 \pm 9.7$ | $198.8 \pm 1.3$ |
| SAC + RAD (GN) | $932.9 \pm 12.9$ | $163.4 \pm 12.6$ | $169.9 \pm 31.0$ |
| Timesteps | 200000 | 50000 | 50000 |

Figure 12: Learning curves of variants of RAD, i.e., random amplitude scaling with a single variable (RAS-S), random amplitude scaling with multivariate variables (RAS-M), and Gaussian noise (GN), on OpenAI Gym. The solid line and shaded regions represent the mean and standard deviation, respectively, across four runs.