[Reviews · NeurIPS 2020]

Review 1

Summary and Contributions: This paper presents Reinforcement learning with Augmented Data (RAD), a plug-and-play module in reinforcement learning to naturally apply data augmentation. The experiments show that using RAD can improve the data efficiency and final performance on both pixel-based environments in Deepmind control suite and state-Applying domain knowledge is quite important to reinforcement learning. RAD can help learning the representation in reinforcement learning, which might be difficult before. This paper also systematically studies data augmentation to pixel-based environments, analyzing which kinds of augmentation are the most important. The learned representation seems to work quite well and can also generalize. The code is released and detailed instructions are attached, which greatly helps reproducibility. based Mujoco environments. RAD can also improve the test-time generalization on pixel-based environments (OpenAI ProcGen). The paper also studies which data augmentation works better and proposes two new data augmentation.

Strengths: Applying domain knowledge is quite important to reinforcement learning. RAD can help learning the representation in reinforcement learning, which might be difficult before. This paper also systematically studies data augmentation to pixel-based environments, analyzing which kinds of augmentation are the most important. The learned representation seems to work quite well and can also generalize. The code is released and detailed instructions are attached, which greatly helps reproducibility.

Weaknesses: The experiments on state-based environments are not convincing enough to me. I have some questions on the reported numbers. It seems that the reported performance of TD3 is quite bad, compared to TD3 paper. For example, TD3 can achieve ~7k rewards in HalfCheetah-v1 after 200k steps according to the TD3 paper, while the reported number here is only 3k. In Walker2d, TD3 paper reports ~2k total rewards after 200k steps, while the number here is < 0. Why is the difference so large? It seems that the error bars are also very large. How many seeds are used to evaluate each algorithm?

Correctness: It's hard to read Figure 12. Maybe smoothing the curves can help. Figure 12 only plots the curves for at most 200k steps, which are not enough to conclude the superior final performance of using RAD. Also the figure prevents me from finding a good gap between SAC and SAC+RAD. L261-263: If multiple variables are used, the relative differences can also be changed. So this argument can't explain why random amplitude scaling can help. Also random amplitude scaling can hurt in some scenarios, e.g., the state is a scalar.

Clarity: In general, this paper is written well. Some minor comments: 1. L259: nothing -> noting 2. As I mentioned above, Figure 12 is difficult to read.

Relation to Prior Work: Authors discuss the difference between RAD and prior work in Section 2 and Appendix A.4.

Reproducibility: Yes

Additional Feedback: POST REBUTTAL: I've read other reviews and authors' rebuttal. The authors answered my questions but I'm still not very convinced why TD3 performs so bad. I'd like to keep my score.


Review 2

Summary and Contributions: This paper investigates data augmentation in RL. Its main contribution is showing that data augmentation is in fact very powerful for RL and could achieve SOTA on some envs.

Strengths: The paper is generally well written and well prepared with extensive related work and code. The empirical study is very important and the results are to some extent significant. The paper should be interesting for all RL researchers.

Weaknesses: While indeed to my knowledge there hasn't been an extensive study of data augmentation in RL (data augmentation, and some other deep learning techniques are curiously missing in RL), the results in this paper are somewhat expected / not that shocking. Data augmentation is very effective on pixel-based RL since they are originally from computer vision. So the SOTA performance of RAD on pixel-RL is more because it helps convolutional neural nets than anything to do with RL. And results on state-based RL (e.g. OpenAI Gym) are not that significant. In the end, it's still that data augmentation helps CNN on images, no matter it's RL or supervised learning. Still, I think it's a good paper and should be published. It has its significance in pixel-based RL itself with many recent works (CURL, DREAMER, PlaNet, etc). In the meantime, due to my comments above, the results are indeed only to some extent significant, so I don't think it warrants a higher score. Even though some augmentation techniques are claimed to be new, the method part itself presents little, if not none, novelty. The novelty could be argued to be demonstrating the potential of data augmentation in RL. One problem on writing: "random translation" is claimed to be one of the two newly introduced augmentations in abstract and introduction, but it's never specifically introduced in Methods (Sec.4), and in Sec.4 the two new techniques proposed are "random amplitude scaling" and "Gaussian noise". This is a paper that readers can grasp in less than five minutes, so there shouldn't be mistakes like this, which must be corrected before publication.

Correctness: Generally yes.

Clarity: Mostly yes.

Relation to Prior Work: Yes.

Reproducibility: Yes

Additional Feedback: ===== post rebuttal ===== Just be sure to correct the confusion I've mentioned.


Review 3

Summary and Contributions: The paper investigates empirically the application of various forms of data augmentation to state-of-the-art reinforcement learning algorithms. Two novel forms of data augmentation are proposed for state-based RL. The empirical verification on a wide variety of RL benchmark problems with continuous control inputs overwhelmingly demonstrates the benefit of data augmentation with respect to reducing significantly the data complexity of the original methods.

Strengths: A major advantage of the proposed data augmentations is that they do not require any modifications to the baseline RL algorithms, and their benefits can be realized by simply appending the original observations with the augmented data.

Weaknesses: Although the proposed general method looks very useful, there does not seem to be a major algorithmic advance in this paper. Most forms of data augmentation used in the evaluation have been known from before, with the exception of the two augmentations specific to state-based RL. Moreover, I am not entirely sure why the data augmentation proposed in this paper works, and this paper does not provide a very good explanation. The main idea is to use data augmentation techniques already used in the area of image classification. However, whereas in image classification problems the class label is invariant to translations, rotations, partial occlusions, some color variations, etc., in RL the correct action in a particular state, respectively when seeing the observation in that state, should not be invariant to them. If there is a one-to-one mapping from state to observation, wouldn't data augmentation make this mapping one-to-many, thus making policy learning harder? I understand that the augmentation is added to the original observation, instead of replacing it, but still I do not understand why it helps. The argument that randomly scaling the true state variable would make the algorithm robust to noise is perhaps acceptable, but all of these simulation environments are deterministic, so this argument should not apply in this verification.

Correctness: They appear to be correct.

Clarity: The paper is written very well.

Relation to Prior Work: The paper does a very good job in reviewing prior work and discussing similarities to and differences from it.

Reproducibility: Yes

Additional Feedback: Minor typos: P.4, L.132: "may perceived" -> "may be perceived" P.6, L.200: "in in" -> "in" --------------------------------------------------------------------------------------------------- Post rebuttal: I think the authors addressed my concerns satisfactorily, although, in the end, there is no theoretical insight as to why data augmentation works, and the main argument for using it are the empirical results presented in the paper. In spite of this, I still think that this paper should be accepted - it proposes a trick that appears to be useful, and does not take a whole lot of effort to implement, so it could be a very useful and practical addition to the state of the art that RL practitioners might appreciate.


Review 4

Summary and Contributions: This paper proposes a reinforcement learning method by incorporating data augmentation into RL. The main contributions that introduce two data augmentations: random translation and random amplitude scaling. Although the author said that it is the first extensive work for RL with data augmentation, the proposed method lack of novelty, because it looks like borrow some augmentation tricks from computer vision and apply it on RL.

Strengths: + The experiments are sufficient and the figures are verifying the data augmentation is working on RL. + The work is well written and easy to read. + Enable a new method for RL with data augmentation. + A structured system. + Reasonable experiments outperforming prior work in qualitative and quantitative evaluations.

Weaknesses: - The method is simple that just like combining data augmentation with RL. - The novelty is not clarified. In summary, the contributions are not enough, which includes some data augmentations. - The work looks nice and well written. However, the technical components adopted in the work are most from the existing ones. It's more a engineering work for a nice application. - We expect to see the challenges as generating images.

Correctness: Yes.

Clarity: Good.

Relation to Prior Work: Yes.

Reproducibility: Yes

Additional Feedback: The feedback addresses my concerns. I hope the authors should provide more details about the novelty in the revision.

[Author Response · NeurIPS 2020]

We thank the reviewers for their feedback. We are encouraged that they found that the empirical study is very important
(R3), the experiments are systematic (R2), reasonable / sufficient (R5), and outperform prior work (R5), and that the
method is simple (R4), easily reproducible with code (R2,R3), and useful for all RL researchers (R3). We address the
reviewers' points of feedback and comments below, and will incorporate all of their feedback.

———————————————————————————— **General Feedback** ————————————————————————————

**Novelty** (R5). "R5: *The novelty is not clarified. In summary, the contributions are not enough, which includes some*
*data augmentations.*" Thank you for your feedback! Our contributions and novelty are stated in the introduction of the
paper. They are (i) first extensive study of data augmentations in the context of RL on various standard benchmarks (as
R2, R3 and R4 mentioned) and (ii) introduction of new data augmentations for both pixel and state-based RL (as R2
and R4 mentioned). It was not obvious before this work that data augmentations were useful in the RL setting, since
they are usually not utilized in pixel-based RL literature. We will be sure to emphasize our contributions in the text to
make these points clearer.

**Simplicity** (R5). "R5: *The method is simple that just like combining data augmentation with RL.*". We agree that the
method is simple, though we view this as a positive. As R4 pointed out, *"A major advantage of the proposed data*
*augmentations is that they do not require any modifications to the baseline RL algorithms, and their benefits can be*
*realized by simply appending the original observations with the augmented data."*

**Generating images** (R5). "R5: *We expect to see the challenges as generating images.*" Thank you for the suggestion!
Generating images / augmentations will be an exciting avenue for future research.

———————————————————————————————— **Text & Logic** ————————————————————————————————

**Explaining why data aug works** (R4). "R4: *Not entirely sure why the data augmentation ... works, and this paper*
*does not provide a very good explanation*". Thank you for pointing out that our explanation could be clearer. Though
we do provide empirical investigations into why data augmentations perform well in Figures 2(b) and 4 of the main
draft, we can certainly improve our explanations. Concretely, we ablate which parts of random crop contribute most in
Figure 6 of the appendix, where we find that translation invariance is the most important aspect of cropping. We can
move this result to the main body and discuss it there to make the explanation clearer.

**Justification on random amplitude scaling** (R4). "R4: *The argument that randomly scaling the true state variable*
*would make the algorithm robust to noise ... should not apply in this verification.*" We remark that there is some
randomness from initial state distribution even though tested simulation environments have deterministic transition
distributions. Because of that, our argument can be valid in our experimental setting. However, we will clarify this in
the final draft. Thank you very much for your pointer.

**Introducing new data augs** (R3). "R3: *Random translation is claimed to be one of the two newly introduced*
*augmentations ... but it's never specifically introduced.*" Thank you for pointing this out to us. We will correct the text
in final draft to avoid confusion.

**Figure 12 legibility**. (R2) "R2: *It's hard to read Figure 12. Maybe smoothing the curves can help. Figure 12 only plots*
*the curves for at most 200k steps.*" We will replace figures with more training timesteps for clarity in the final draft.

**Random amplitude scaling with multiple variables**. (R2) "R2: *L261-263: If multiple variables are used, the relative*
*differences can also be changed.*" That's a good point. As you mentioned, the relative differences can be changed in
the case of random amplitude scaling with multiple variables. Because of that, random amplitude with a single scalar
achieves the better performance on most environments. We will clarify this part in the final draft.

———————————————————————————————— **Experiments** ————————————————————————————————

**Results on state-based RL** (R3). "R3: *Results on state-based RL ... are not that significant.*" We believe that our
experimental results on OpenAI Gym are extensive and demonstrate the strength of RAD in that (a) we consider strong
baselines, such as POPLIN and PETS, and (b) our method provides large gains in complex environments like Walker.

**Performance of TD3 algorithms** (R2). "R2: *The experiments on state-based environments are not convincing*
*... performance of TD3 is quite bad, compared to TD3 paper.*" We emphasize that the version of OpenAI Gym
environments is different from TD3 paper (we used the setups of POPLIN, which is published in ICLR 2020), and we
took the best reported performance for TD3 in POPLIN. We also checked out that similar scores can be reproduced by
the official codebase from the POPLIN paper (e.g., 3273.4 on Cheetah and -447.3 on Walker using 10 random seeds).
To clarify this concern, we will update the scores on all environments using 10 random seeds.

**Application specific** (R5). "R5: *More a engineering work for a nice application*" We show state-of-the-art results on
common RL benchmarks like DeepMind control, ProcGen, and OpenAI Gym, which are standard benchmarks used to
study RL by a suite of other general purpose RL algorithms (CURL, SLAC, Dreamer, PlaNet, PPO, POPLIN, PETS).

[Meta-Review · NeurIPS 2020]

The paper investigates various data augmentation techniques in the context of RL, and shows that they lead to improved performance. The method is simple and can be applied to different RL algorithms. One may argue that the algorithmic contribution is not majorly novel, but the simplicity of the method and the improvement in the performance, as well as the unanimous favourable opinion of the reviewers, would be good enough justification to recommend acceptance of this work. I encourage the authors to consider the comments of the reviewers in revising their paper. I would also like to ask the authors to increase the number of runs/seeds in some of their experiments from 3 or 4 to a larger number (10+). Currently, the difference between methods may not be statistically significant in some cases. I believe having a spotlight is suitable for this work, as many RL practitioner can benefit from knowing about the positive effect of data augmentation in RL.